# Assessment of Life Quality in Children with Dysphonia Using Modified Pediatric Voice-Related Quality of Life Questionnaire in Serbia

**DOI:** 10.3390/children10010125

**Published:** 2023-01-06

**Authors:** Jasmina Stojanovic, Mila Veselinovic, Milica Jevtic, Marina Jovanovic, Dusan Nikolic, Jovana Kuzmanovic Pficer, Emilija Zivkovic-Marinkov, Nenad Relic

**Affiliations:** 1Otolaryngology Clinic, University Clinical Center, 34000 Kragujevac, Serbia; 2Department of Otorhinolaryngology, Faculty of Medical Sciences, University of Kragujevac, 34000 Kragujevac, Serbia; 3Department of Special Education and Rehabilitation, Faculty of Medicine, University of Novi Sad, 21137 Novi Sad, Serbia; 4Clinic for Otorhinolaryngology and Head and Neck Surgery, University Clinical Center Vojvodina, 21137 Novi Sad, Serbia; 5Department of Chemical Engineering—Pharmaceutical Engineering, Faculty of Technology and Metallurgy, University of Belgrade, 11000 Belgrade, Serbia; 6Department of Medical Statistics and Informatics, School of Dental Medicine, University of Belgrade, 11000 Belgrade, Serbia; 7ENT Clinic, University Clinical Center, 18000 Niš, Serbia

**Keywords:** children, dysphonia, quality of life, modified pediatric voice-related quality of life questionnaire

## Abstract

(1) Background: Hoarseness is not uncommon in children, especially at school age, as communication with peers is intensified. It is caused by improper use or overuse of the vocal apparatus. (2) Methods: The study included 85 hoarse children aged 6–12 (study group) and 240 healthy children (control group) of the same age. The study group underwent a detailed medical history, phoniatric examination, larynx fiber endoscopy, allergy treatment and the Pediatric Voice-Related Quality of Life questionnaire, modified by Jasmina Stojanovic. (3) Results: Our modified questionnaire revealed the significance of parental perception of a voice disorder in a child after organized activities. Using our modified questionnaire, we were able to determine the most frequent form of a voice disorder in children—speaking too loudly—is often neglected by the environment and can lead to an overall lower life quality. (4) Conclusions: As the presence of hoarseness impairs the quality of life in the pediatric population, awareness of a voice disorder must be recognized and treated on time to overcome the possible side effects on a child’s psychological and emotional development.

## 1. Introduction

The concept of quality of life was initially considered a useful addition to the traditional concepts of health and functional status, whereas today’s understanding depicts the quality of life as a necessity in modern medicine [1].

Pediatric dysphonia is a disorder that manifests itself in a wide range of symptoms and signs, from mild hoarseness to a complete inability to communicate. It is characterized by an altered voice volume, pitch or vocal effort, which reduces voice quality and makes communication difficult [2]. According to epidemiological studies, the prevalence of dysphonia in children ranges between 6 and 38%, and it can negatively affect a child’s life in terms of their general health, communication efficiency, social and educational development, as well as participation in group activities in kindergarten and school [3]. Most children’s vocal disorders occur between the ages of 5 and 10. The most common organic cause of pediatric dysphonia is vocal cord nodules. Pediatric dysphonia is often underestimated. This is a serious problem because it can lead to chronic voice changes that can limit a child’s success in school but also their social and professional opportunities in later life [4,5].

In the past two decades, several questionnaires have been designed and created to assess the quality of life of patients with vocal problems. The most used and valid are the following questionnaires: Voice Handicap Index, Voice Outcomes Survey, and Voice-Related Quality of Life, originally designed for use in adult patients with vocal disorders [6]. Normative data for a healthy population have also been published [7]. Questionnaires assessing the quality of life in the pediatric population are completed by parents of children with dysphonia [8]. The Pediatric Voice-Related Quality of Life questionnaire is valid for assessing the quality of life of hoarse children [9].

In Serbia, children’s hoarseness is still not sufficiently recognized as a health problem, as there were no previous studies concerning this subject. This leads to numerous consequences, not only in preschool and school age but potentially in adulthood as well.

The aim of our study is to investigate the quality of life in hoarse children in Serbia aged 6 to 12 years, using a modified Mark Boseley Pediatric Voice-Related Quality of Life questionnaire by Jasmina Stojanović (MPVRQOL).

## 2. Materials and Methods

### 2.1. Questionnaire Subministration

A prospective case-control study that involved hoarse pediatric patients was performed at the Phoniatric Department, Otorhinolaryngology, University Clinical Center of Kragujevac. The study included 85 hoarse children, 57 male and 28 female, (study group) aged 6–12, and the control group comprised 240 healthy (118 male and 122 female) children of the same age from two primary schools in two Serbian towns, from November 2020 to January 2022. The children were matched by age and gender, in a case-control ratio of 1:3. Children with previously diagnosed dyslalia or any other speech disorder, as well as a history of transient hoarseness lasting longer than two weeks at any point, were excluded from the study. All subjects and caregivers gave their informed consent for participation in the study, and the ethical guidelines of the Declaration of Helsinki were followed during the study.

### 2.2. Patients’ Assessment

The healthy children were chosen by a method of random sampling in two primary schools. Detailed medical histories of the study group were obtained, and the children underwent a thorough medical examination, including phoniatric examination, fiber endoscopy larynx, allergy treatment, audiological treatment, pulmonary treatment, and a multi-dimensional computer software voice analysis using Jitter, Shimmer and standard deviation as notable parameters. Jitter and Shimmer are used as markers of variations in the fundamental voice frequency caused by an irregular vocal fold vibration. Jitter is used to measure the variations in a voice frequency in an individual, whereas Shimmer is used to measure the amplitude of sound waves indicative of a voice’s emission [10].

### 2.3. MPVRQOL Questionnaire

The study applied MPVRQOL to be completed by the parents of hoarse children and parents of healthy children as a valid instrument for assessing life quality. The questionnaire we used in our research comprises 11 questions, 7 of which refer to the physical aspect of the quality of life of hoarse children and 4 to the socio-economic aspect. The questionnaire was translated from English to Serbian, culturally adapted for each item, and modified by the first author by adding another question in regards to the social aspect of life quality (modification by Jasmina Stojanovic, with the consent of Mr. Boseley. The purpose of the added question (“My child has a hoarse voice after organized activities (childrens’ parties, birthday parties, sports games)”) was to examine further a social aspect of child hoarseness. In order to ensure the accuracy of the translated questionnaire, a back-translation to English was completed by another translator who was unfamiliar with the original version. The questionnaires were administered by the first author through face-to-face interviews. The PVRQOL is an 11-item instrument designed to measure VRQOL (voice-related quality of life) and is adapted from the adult and pediatric VRQOL instruments. The scores of the instrument were normalized to a scale of 100 for ease of interpretation. A score of 100 represents the highest QOL, which meant that the parents perceived no problems with their child’s voice, no limitations on voice function, and no adverse social or emotional effects attributable to their child’s voice quality. The hoarse children had already been given a diagnosis during the first-time examination and adequate therapy. However, the study did not include the treatment results, as the goal of the study was the assessment of the quality of life of hoarse children during the first examination. The questionnaire used, which was translated into English, is shown in the Appendix A.

### 2.4. Statistical Analysis

The complete statistic analysis was performed using the IBM SPSS Statistics 22.0 computer program. All continuous variables (age, scores of scales) are shown in the form of the mean ± standard deviation, while the categorical variables (gender, marital status) are shown with the percentage of certain category frequency. For the categorical variables, the statistical significance of differences was examined by a Chi-square test. For the continuous variables, after analyzing the distribution of the data using the Kolmogorov–Smirnov test, a Student’s *t*-test or Mann–Whitney U test was used, where appropriate. The correlation between the two continuous variables was examined by Pearson’s linear correlation or Spearman’s rank correlation.

## 3. Results

The study included 325 children aged between 6 and 12 years, of whom 85 were hoarse (study group), and the remaining 240 had no vocal problems (control group). All children included in the study had normal psychomotor development, as well as an orderly finding of tone audiometry. In the examined group, there were 57 boys and 28 girls, while there were 118 boys and 122 girls in the control group. (Table 1). The average age of children in the examined group was 8.60 ± 1.941 years, and in the control group, 8.82 ± 1.61 years. There is a statistically very significant difference between the study and control groups in relation to gender (*p* < 0.01); however, there is no statistically significant difference between these two groups in relation to age (*p* > 0.05).

When it comes to the number of children per family of examined subjects, there were 45 children from a family with two children (*n* = 45, 52.94%), 25 children from a family with one child (*n* = 25, 29.41%), and 15 children from a family with three children (*n* = 15; 17.65%). There were no children from a family with more than three children in this group. Vocal nodules were diagnosed in 58 hoarse children, while the remaining 27 children in the study group were diagnosed with laryngeal muscle tension disorder (hyperkinetic dysphonia) (Table 1). Clinical examination and fiber laryngoscopy established that all children in the examined group had insufficient glottis occlusion with noise in the voice and pronounced external signs of hyperkinesia. Out of the total number of children in this group, 11 (12.9%) had allergic rhinitis and 6 (7.1%) had bronchial asthma. Statistical analysis of the hoarse children’s voice quality data, obtained by a computer multi-dimensional voice analysis, found that there was a statistically very significant (*p* < 0.01) negative correlation between the Jitter value and the value of the socio-emotional domain and a statistically significant (*p* < 0.05) negative correlation between the value Jitter and total domain values. There was no statistically significant correlation (*p* > 0.05) between the Jitter values and physical domain values, as well as the Shimmer values and values of all three domains (Table 2).

The average values of the answers to the questions from the questionnaire given by the parents of children from the examined and control groups, as well as the results of the statistical analysis of these answers, are summarized in Table 3. There is a statistically significant difference (*p* < 0.05) in the average values of the answers to questions number 1, 7 and 11. Moreover, there is a statistically very significant difference (*p* < 0.01) in the average values of the responses related to the value of the physical and total domain of the MPVRQOL scale, while there is no statistically significant difference (*p* > 0.05) in the average values of their socio-emotional domain scale.

Analyzing the mean value of the responses on the scales between the examined and control groups in relation to gender, we found that, in the examined group, there was a statistically significant difference in the responses for each domain separately (Table 4).

This difference is statistically significant when it comes to the mean values of the responses related to the value of the physical, socio-economic and total domain of the MPVRQOL scale (*p* < 0.05). Additionally, there is a statistically very significant (*p* < 0.01) difference in the average value of responses related to the value of the physical and total domain between the examined and control group by gender.

The largest number of hoarse children—63 (74.1%) received a grade of 20 on the questionnaire “fair to good”, while the largest number of children from the control group—173 (72.1%) received a grade of 10, noted as “excellent” (Table 5). Four children from the control group (1.7%) received a grade of 30, “poor to fair”, on the questionnaire. It should be noted that there is a statistically very significant (*p* < 0.01) difference in the grades obtained by hoarse children and children from the control group.

## 4. Discussion

Dysphonia represents the impairment of voice production, clinically represented as hoarseness, which is a symptom of distorted voice quality [3]. As dysphonia can be a natural part of the aging process, it can also be a symptom of an unrecognized underlying condition [12]. When it comes to dysphonia in adults, a complete phoniatric examination is performed, outlining the cause and preferred way of treatment [13]. On the other hand, when dealing with pediatric dysphonic abnormalities, whatever the prevalence, its occurrence and causes are not being addressed enough during childhood development [14]. Pediatric dysphonia represents a wide-ranging spectrum of voice abnormalities in a child, with symptoms from hoarseness to the incapability to communicate normally [15]. Voice disorders in children, although not so uncommon, have been neglected in the past, discarding them as transient and unimportant [16]. Nowadays, it is known that, in addition to various functional problems with the vocal apparatus, voice disorders in a child might contribute to hindered educational and psychological development, therefore decreasing the overall quality of life in these children [17].

The main goal of this study was to elucidate the impact of voice disorders on children and their quality of life. We used a modified version of MPVRQoL to determine the quality of life of a Serbian children’s population. The MPVRQOL scale was found to have high internal consistency with Cronbach’s alpha = 0.944, which means that the reliability of the scale is excellent. This study is a continuation of our previously published data, as we offer new insights on the problems discussed, aiming to raise awareness of the complex nature of pediatric dysphonia and its impacts on the development of children [14].

Multi-dimensional voice analysis of hoarse children, using the well-established parameters Jitter and Shimmer, showed a significant inverse correlation between the values of Jitter and the total domain score, with emphasis placed on the socio-emotional domain. As Jitter is used to measure a sound wave’s frequency variation from cycle to cycle, our results imply that one of the underlying problems in hoarse children is having no control over the vibration of the vocal cords, which hinders the physiological production of the voice [18]. This consequently affects the socio-emotional domain of hoarse children, which might lead to psychological hindrances during childhood development [19]. On the other hand, as expected, the values of both parameters (Jitter and Shimmer) had no correlation with the physical domain. In addition, Shimmer was no different in hoarse children in comparison to the control group in every domain. Since Shimmer measures the reduction of glottal resistance during voice production, hoarse children have physiological-level glottal resistance in the majority of the cases. As it is well known, glottal resistance is reduced when mass lesions on the vocal cords are present and is usually followed by breathiness [20]. This is in line with our results of local findings in the larynx of hoarse children since all hoarse children have signs of hyperkinetic dysphonia, and only some of them have incipient nodules [21].

Given the results concerning the social and economic aspects of the examined children, there was no statistically significant difference, meaning that the social and economic aspects had no impact on the hoarseness of children. However, it cannot be disregarded that vocally impaired children could potentially have difficulties during communication with their peers, with mild to severe psychological consequences [22]. Additionally, whatever the social background in which a vocally impaired child evolves, listeners’ attitudes toward these children can be less favorable. This observation, from a study by Ma et al. [23], is especially important when it comes to educators. Having this in mind, the emphasis of the prevention and care of pediatric vocal disorders is an early acknowledgment of the problem [24]. It is not uncommon, however, that a voice disorder in a child remains unnoticed by parents and teachers, especially when the child is otherwise healthy. Hoarseness in a child is thought to be of a “cosmetic” nature, or it is thought to be transitory. It must be taken into account that, even if transitory, children with an impaired voice might have permanent unwanted outcomes regarding their psychological development [25].

According to our modified questionnaire, there was a significant (Table 1; *p* < 0.01) difference between the questioned and control groups when asked whether the child speaks too loudly. This result is in line with our previously published data [14]. This result reflects the awareness of a problem for parents, which is considered one of the main steps towards the resolution of a voice disorder in a child [7]. In line with this data, there are many studies showing that the most common form of vocal misuse in children is speaking too loudly [20,26,27,28]. Alongside that, the significant difference between the examined and control groups was also noted when parents were asked to determine whether their child’s voice was hoarse after participating in organized activities with peers, such as sports games, parties, etc. We added this question to MPVRQOL since we hypothesized that during intense social activities, vocal misuse could be pronounced since it is in a child’s nature to misuse their voice during a gathering of peers. There are studies that imply that, psychologically, in a child, this can be explained by a desire to override other children [21]. Our results support this observation. Another attribute worth mentioning is that vocally impaired children have, according to our results, significantly more difficulties when it comes to dealing with scholarly obligations. As mentioned before, even though vocal disorders in children might seem mild, they subtly interfere with children’s everyday life at an age where social and behavioral patterns are implemented. Even though there are studies that imply psychological disturbances might occur later in life for a vocally impaired child, this notion is not emphasized enough [29,30].

When we analyzed our scale, according to gender and the examined domains (global, physical and socio-emotional domains), we found that there is a significant difference in every domain. Interestingly, female children had significantly greater scores in both the examined and control groups (Table 4.) Additionally, the values of every domain score were significant in both the dysphonic male and female children in comparison to the control groups of both gender**s**, with the exception of the socio-emotional domain of male participants (Table 4). Having taken the scores of different domains into consideration, it is clear that male children have more tendencies towards the misuse of their voice during school age. This observation is not infrequent—many studies have highlighted the role of gender during voice disorders in children [31,32,33,34], yet the overall awareness of voice misuse in children has not been raised.

## 5. Conclusions

There must be more raised awareness towards the recognition of a vocal disorder, especially by a parent or educators. Raising awareness of a vocal disorder in a child will certainly cut down the period for a first visit to an ENT specialist and speech therapist and therefore impede the adoption of unwanted speech mechanisms throughout life. More importantly, as implicated in our results, emphasis must be placed on vocal disorders in male children, as dysphonia in male children can often be disregarded due to changes in the voice as secondary sexual characteristics. Additionally, given the results of using a modified questionnaire, it is important to note that vocal disorders in a male child could be disguised as transitory hoarseness after organized activities, trouble with writing homework, or sometimes even not being able to speak loudly.

## Figures and Tables

**Table 1 children-10-00125-t001:** Demographic characteristics of children involved in the study (*n* = 325).

	Number (%)		*p*-Value
	Dysphonic Subjects 85 (26.2)	Control Subjects 240 (73.8)	
Gender ^1^			*p* = 0.004 *
Male	57 (67.1)	118 (49.2)	
Female	28 (32.9)	122 (50.8)	
Age; M ± SD ^2^	8.60 ± 1.941	8.82 ± 1.61	*p* = 0.159
Age; M ± SD ^2^			
Male	8.58 ± 2.01	8.79 ± 1.65	*p* = 0.625
Female	8.64 ± 1.83	8.84 ± 1.58	*p* = 0.589
Number of children per family:			
1 child	25 (29.41)		
2 children	45 (52.94)		
3 children	15 (17.65)		
Clinical disorder			
Muscle tension	27 (31.8)		
Vocal nodules	58 (68.2)		

^1^ Chi-square test; ^2^ Mann–Whitney test. * Statistically significant.

**Table 2 children-10-00125-t002:** Correlation between the general, socio-emotional and physical scores of the modified PVRQOL scale and the vocal quality evaluation in the study group [11].

	Vocal Quality Evaluation		
	Jitter	Shimmer	SD
Physical domain score	r = −0.161; *p* = 0.142	r = 0.005; *p* = 0.966	r = −0.021; *p* = 0.850
Socio-emotional domain score	r = −0.281; *p* = 0.009 *	r = −0.207; *p* = 0.062	r = −0.026; *p* = 0.810
Global domain score	r = −0.217; *p* = 0.046 *	r = −0.057; *p* = 0.613	r = −0.030; *p* = 0.788

Pearson’s correlation (***** statistically significant).

**Table 3 children-10-00125-t003:** Average values of all the scale scores in the two groups.

	Dysphonic Subjects		Control Subjects		*p*-Values ^1^
	Mean	SD	Mean	SD	
Question 1	3.35	1.78	2.05	1.82	*p* < 0.001 *
Question 2	1.82	1.53	1.90	1.78	*p* = 0.946
Question 3	1.67	1.48	1.96	1.88	*p* = 0.902
Question 4	2.29	2.06	2.10	1.82	*p* = 0.936
Question 5	1.73	1.53	2.15	2.04	*p* = 0.329
Question 6	1.88	1.86	2.04	1.97	*p* = 0.474
Question 7	2.14	1.98	1.88	1.86	*p* = 0.042 *
Question 8	2.02	2.01	2.03	1.98	*p* = 0.894
Question 9	1.82	1.47	2.10	1.98	*p* = 0.342
Question 10	1.52	1.48	2.03	2.02	*p* = 0.072
Question 11	3.02	1.08	1.69	0.94	*p* < 0.001 *
Physical domain score	11.54	3.03	8.56	2.16	*p* < 0.001 *
Socio-emotional domain score	4.56	1.13	4.48	1.17	*p* = 0.545
Global domain score	16.11	3.79	13.04	3.06	*p* < 0.001 *

^1^ Mann–Whitney test (***** Statistically significant).

**Table 4 children-10-00125-t004:** Gender differences in the scale scores in the two groups.

		Dysphonic Subjects		Control Subjects		*p*-Values ^1^
		Mean	SD	Mean	SD	
Physical domain score	Male	10.79	2.62	8.41	1.97	*p* < 0.001 *
	Female	13.07	3.28	8.70	2.33	*p* < 0.001 *
	*p*-values *	*p* = 0.003 *		*p* = 0.190		
Socio-emotional domain score	Male	4.40	1.10	4.49	1.19	*p* = 0.348
	Female	4.89	1.13	4.47	1.15	*p* = 0.016 *
	*p*-values *	*p* = 0.011 *		*p* = 0.964		
Global domain score	Male	15.19	3.32	12.90	2.91	*p* < 0.001 *
	Female	17.96	4.04	13.17	3.21	*p* < 0.001 *
	*p*-values *	*p* = 0.002 *		*p* = 0.254		

^1^ Mann–Whitney test (***** Statistically significant).

**Table 5 children-10-00125-t005:** Questionnaire scores according to group of subjects.

Quiz Score	Questionnaire Score	Dysphonic Subjects	Control Subjects	*p*-Value *
10	100 (excellent)	21 (24.7%)	173 (72.1%)	*p* < 0.001
20	75 (fair to good)	63 (74.1%)	63 (26.3%)	
30	50 (poor to fair)	1 (1.2%)	4 (1.7%)	
40	25 (poor)	-	-	
50	0 (worst possible)	-	-	

* Chi-square test.

## Data Availability

The data presented in this study are available on request from the corresponding author.

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
