# Peer review of "Assessment of Life Quality in Children with Dysphonia Using Modified Pediatric Voice-Related Quality of Life Questionnaire in Serbia"

_children, 2023, doi:10.3390/children10010125_

Round 1

Reviewer 1 Report

The quality of life in children with dysphonia using modified Pediatric Voice Related Quality of Life  questionnaire

The topic is interesting,  a great work has been done.

I have only few remarks.

The most used and valid are the following questionnaires: Voice Handicap Index, Voice Outcomes Survey, Voice Related  Quality of Life, originally designed for use in adult patients with vocal disorders.  References are missing

In Serbia, children's hoarseness is still not sufficiently recognized as a health problem  Some reference?

Mark Boseley Pediatric Voice Related Quality of Life questionnaire by Jasmina Stojanović (MPVRQOL ) The reference is missing

Methods

All subjects gave their informed consent.  I imagine you mean also the parents or  caregivers..

I suggest  to split out the section in more “sub sections”. Just an example: 2.1 questionnaire subministration 2.2 statistics, etc .In this way the reader can find immediately the information he need

 multidimensional computer software voice analysis (Jitter, Shimmer, and Standard Deviation). I would briefly explain whate are  jitter and shimmer because not all pediatricians know these parameters, being this journal  read not only by ENT specialists

Results

The questionnaire we used in our research consists of 11 questions, 7 of which refer to the physical aspect of the quality of life of hoarse children, and 4 to the socio-economic aspect. This sentence shoul be inserted in methods section

Lines 165-186:this section is just a list of results that are indicated in the corresponding table… I suggets to report the questionnaire in the methods section or as appendix. In my opinion, the related  paragraph  of results coud be shortened and be more easily  readable

Discussion

one thing is certain I woul eliminate it…

In our study, we used Mark Boseley Pedatric Voice Related Quality of  Life (MPVRQoL) which we modified, with the consent of the author, by addition of one  single question in regards to social aspect of life quality (modification by Jasmina Stojanovic). This part shoul be inserted in methods section. Which question has been added ?  why ? It should be better explained in methods section..

Author Response

The topic is interesting,  a great work has been done.

I have only few remarks.

The most used and valid are the following questionnaires: Voice Handicap Index, Voice Outcomes Survey, Voice Related  Quality of Life, originally designed for use in adult patients with vocal disorders.  References are missing

In Serbia, children's hoarseness is still not sufficiently recognized as a health problem  Some reference?

Mark Boseley Pediatric Voice Related Quality of Life questionnaire by Jasmina Stojanović (MPVRQOL ) The reference is missing

Thank you for your comments. We have now added the reference about most used and valid questionnaires. However, when it comes to any study of importance of pediatric dysphonia in Serbia, we were not able to address it accordingly, since there were no previous studies about the subject, other than our group.

 Methods

All subjects gave their informed consent.  I imagine you mean also the parents or  caregivers..I suggest  to split out the section in more “sub sections”. Just an example: 2.1 questionnaire subministration 2.2 statistics, etc .In this way the reader can find immediately the information he need  multidimensional computer software voice analysis (Jitter, Shimmer, and Standard Deviation). I would briefly explain whate are  jitter and shimmer because not all pediatricians know these parameters, being this journal  read not only by ENT specialists

We are sorry for unprecised vocabulary. We now changed the sentence with “All caregivers gave their informed consent. Also, as suggested, we split out the Methods section into following sub-sections: Questionnaire subministration, Patient assessment, MPVRQOL questionnaire, Statistical analysis. We have also more thoroughly described multidimensional computer software analysis.

Results

The questionnaire we used in our research consists of 11 questions, 7 of which refer to the physical aspect of the quality of life of hoarse children, and 4 to the socio-economic aspect. This sentence shoul be inserted in methods section Lines 165-186:this section is just a list of results that are indicated in the corresponding table… I suggets to report the questionnaire in the methods section or as appendix. In my opinion, the related  paragraph  of results coud be shortened and be more easily  readable

We thank you for the constructive commentary. We have now, as suggested, inserted the sentence in Methods section. Also, we have now reported the questionnaire as an Appendix. Hence, related paragraph was shortened accordingly.

Discussion

one thing is certain I woul eliminate it…

In our study, we used Mark Boseley Pedatric Voice Related Quality of  Life (MPVRQoL) which we modified, with the consent of the author, by addition of one  single question in regards to social aspect of life quality (modification by Jasmina Stojanovic). This part shoul be inserted in methods section. Which question has been added ?  why ? It should be better explained in methods section..

We have now inserted the sentence in Methods section as proposed. We have also more thoroughly explained which question and the underlying reason it has been added.

Reviewer 2 Report

Dear Authors,

Thank you for submitting your article entitled " The quality of life in children with dysphonia using modified Pediatric Voice Related Quality of Life questionnaire". This study is an original study that examines the quality of life in hoarse children in Serbia, using a modified questionnaire.

Overall, this is well written and informative manuscript. However, it needs some important points to be rectified.

I summarize the following

1)    ABSTRACT

a.     Overall, the abstract is well written. However, it should be reformed based on the comments mentioned below.

2)    MAIN PAPER

a.     TITLE

I believe that the title should include a term such as “evaluation or assessment” to better support the aim of this study. Additionally, since the study is conducted in Serbia, it would be great to be included in the title.

b.    INTRODUCTION

Overall, this section is well written. However, there are some points that should be addressed. Specifically:

(1)   The content of the first paragraph (“The concept […] patient’s health [1].”) should be kept to a minimum. I strongly believe that the essence of the quality of life can be provided in a single sentence, since its’ content is known from 1998 (as reflected by reference 1).

(2)   The content of the second paragraph (“Pediatric dysphonia […] in its later life [4,5].”) should be reduced in size since, as reported, is relatively large. I strongly believe that the authors should briefly present the most crucial aspects of dysphonia and general information (such as the fact that dysphonia is more common in boys (“Dysphonia […] excessively.”) should be kept to a minimum.

(3)   As already mentioned above, again the content of the third paragraph (“In the past […] dysphonia [7].”) should be significantly reduced. I strongly believe that general information such as the most used questionnaires should be erased or at least substituted by references.

(4)   As per the fourth paragraph (“The Pediatric […] estimated [8].”), I strongly believe that it should be merged with the third one since, their content is similar. Additionally, a brief reference of the aspects of the questionnaire that was used should be reported in the introduction, since they will be meticulously provided in the Materials and Methods.

(5)   The fifth paragraph (“In Serbia, […] well.”) should be supported by a proper amount or references. Otherwise, it should be erased as non-evidence-based speculation.

(6)   As per the sixth paragraph (“The aim […] children [9].”) the following should be considered:

(a)   The term “work” should be substituted by a more appropriate one (i.e., “study”).

(b)  The term “determine” should be substituted by a more appropriate one (i.e., “investigate”) since no definite conclusions can be drawn as mentioned in the results section below.

(c)   The sentences “This study […] of children [9].” should be moved in the Discussion since, to the best of my understanding are not relevant to the study’s novelty, and the primary or secondary goals.

c.     MATERIALS AND METHODS

Overall, this is a well informative section. However, there are several shortcomings that should be amended and are specified below.

(1)   The methodology that was used needs to be clarified in a greater extend since, as reported it is confusing. Specifically, the following should be explained in a greater extend:

(a)   It is reported that this study is a cohort (prospective, even if mentioned). Yet, to the best of my understanding, it uses cases and controls and is in fact a “prospective case-control study”. Therefore, if my understanding is correct, the relevant changes should be performed.

(b)  As I can speculate, both the hoarse and the healthy children derived from the very same two Serbian primary schools. Yet, in text, it is reported that the controls derived from elementary schools (line 92). Therefore, the relevant changes should be performed.

(c)   Additional information about the selection of controls (i.e., the case-control ratio, matching etc.) and the timeframe of the study should be reported.

(d)  All the variables that were recorded should be provided. There should be no variables that are not reported in the Materials and Methods and are initially provided in other sections (such as the number of children).

(e)   The rationale of modifying the questionnaire should be also reported. The authors should also provide data about the validity of the questionnaire after the modification.

(f)    The section related to the informed consent should be modified accordingly if the parents gave the informed consent.

(g)   The assessment of the continuous variables normality (normality tests) should be provided to report the results in mean and standard deviation values. If not normally distributed other measures of central tendency and dispersion should be used. For instance, to the best of my understanding, I guess that age should be not normally distributed. Based on that, the authors should also be cautious about the tests that will be used if, not normally distributed data is used.

(2)   The last sentence (The MPVRQOL […] should be moved in the discussion since it is not related to the methodology used to perform this study.

(3)   Finally, I strongly believe that a sample of the modified version of the questionnaire should be included as a figure since, to the best of my understanding, it serves as a novel modality that can be used by other authors interested in this field of research. 

d.    RESULTS

(1)   The content of the first paragraph (“The study […] (p>0.05)”.) should be reduced in size and a Table where all patients’ characteristics will be synopsized needs to be created. In addition, clauses such as “There were twice as many boys as girls […].” should be modified, to report only the findings of the study without commentaries.

(2)   I strongly believe that the term “gender” should be replaced by “sex”. Please refer to the following: https://www.coe.int/en/web/gender-matters/sex-and-gender

(3)   As mentioned in the Materials and Methods, all variables that are reported in the results (such as the largest number of children or the content of table 2) should be also included in the relevant section.

(4)   The strength of the correlation coefficient as reported in lines 155-161 (“Statistical […] (Table 3).”) and in Table 3 should be explained in a greater extend using a reference where it is defined. 

(5)   The content of the paragraph (“The questionnaire we used […] hoarse children.”) as stated in lines 165-186 should be significantly reduced in size. Since the relevant data is reported in a table, only a synopsis should be provided in the results and if needed, commentary can be performed in the discussion. Moreover, the authors should not repeat the structure of the questionnaire. 

(6)   The content of the paragraph (“The largest number […] control group.”) should be significantly reduced in size. 

e.     DISCUSSION

Overall, I believe that it is an extremely well written and informative section. My biggest concern is only related to its length. Hence, I suggest being reduced as much as possible. Moreover, the study’s strengths and limitations, including the potential confounding and the bias assessment should be reported and discussed. 

As per the conclusion, I strongly believe that a separate section should be created. Yet, the text of this section should be reduced and, the authors should be more specific on addressing the novelty of this study as well as of the findings. 

f.      TABLES

i.      I strongly believe that the content of Table 2 can be included in the newly created table 1 as proposed above since as given is too small and all displayed information can be reported in text. 

ii.     As per the Table 4, I strongly believe that each question should be written full-text especially if the questionnaire will not be provided as a figure.

Author Response

Thank you for submitting your article entitled " The quality of life in children with dysphonia using modified Pediatric Voice Related Quality of Life questionnaire". This study is an original study that examines the quality of life in hoarse children in Serbia, using a modified questionnaire. Overall, this is well written and informative manuscript. However, it needs some important points to be rectified. I summarize the following: 

1)    ABSTRACT

  1. Overall, the abstract is well written. However, it should be reformed based on the comments mentioned below.

Thank you for your commentary. According to the proposition, abstract is now reformed to suit revised manuscript more.

2)    MAIN PAPER

  1. TITLE

I believe that the title should include a term such as “evaluation or assessment” to better support the aim of this study. Additionally, since the study is conducted in Serbia, it would be great to be included in the title.

Thank you for the proposition. We have now altered the title and the manuscript is now called: “Assessment of life quality in children with dysphonia using modified Pediatric Voice Related Quality of Life questionnaire in Serbia”. 

  1. INTRODUCTION

Overall, this section is well written. However, there are some points that should be addressed. Specifically:

(1)   The content of the first paragraph (“The concept […] patient’s health [1].”) should be kept to a minimum. I strongly believe that the essence of the quality of life can be provided in a single sentence, since its’ content is known from 1998 (as reflected by reference 1).

We have shortened the first paragraph, and depicted the importance of life quality in hoarseness making it more concise.

(2)   The content of the second paragraph (“Pediatric dysphonia […] in its later life [4,5].”) should be reduced in size since, as reported, is relatively large. I strongly believe that the authors should briefly present the most crucial aspects of dysphonia and general information (such as the fact that dysphonia is more common in boys (“Dysphonia […] excessively.”) should be kept to a minimum

Thank you for the suggestion. We have also shortened the mentioned paragraph.

(3)   As already mentioned above, again the content of the third paragraph (“In the past […] dysphonia [7].”) should be significantly reduced. I strongly believe that general information such as the most used questionnaires should be erased or at least substituted by references.

We have also shortened the paragraph in question, and erased some of the sentences, making it more precise, yet sufficient.

4)   As per the fourth paragraph (“The Pediatric […] estimated [8].”), I strongly believe that it should be merged with the third one since, their content is similar. Additionally, a brief reference of the aspects of the questionnaire that was used should be reported in the introduction, since they will be meticulously provided in the Materials and Methods.

We have corrected the paragraph mentioed above, and also shortened our refference of aspects in questionnaire used.

(5)   The fifth paragraph (“In Serbia, […] well.”) should be supported by a proper amount or references. Otherwise, it should be erased as non-evidence-based speculation

Thank you for your observation. We unfortunetely have no refferences to disclose, as our group is the first to engage in pediatric hoarseness.

(6)   As per the sixth paragraph (“The aim […] children [9].”) the following should be considered:

(a)   The term “work” should be substituted by a more appropriate one (i.e., “study”).

(b)  The term “determine” should be substituted by a more appropriate one (i.e., “investigate”) since no definite conclusions can be drawn as mentioned in the results section below.

(c)   The sentences “This study […] of children [9].” should be moved in the Discussion since, to the best of my understanding are not relevant to the study’s novelty, and the primary or secondary goals.

We have substituted the proposed words, and we have also moved the suggested sentence in Disscusion section.

  1. MATERIALS AND METHODS

Overall, this is a well informative section. However, there are several shortcomings that should be amended and are specified below.

(1)   The methodology that was used needs to be clarified in a greater extend since, as reported it is confusing. Specifically, the following should be explained in a greater extend:

(a)   It is reported that this study is a cohort (prospective, even if mentioned). Yet, to the best of my understanding, it uses cases and controls and is in fact a “prospective case-control study”. Therefore, if my understanding is correct, the relevant changes should be performed.

We are sorry for the uninattentional mistake. We have now switched the term “prospective” with the term “prospective case – control study”

(b)  As I can speculate, both the hoarse and the healthy children derived from the very same two Serbian primary schools. Yet, in text, it is reported that the controls derived from elementary schools (line 92). Therefore, the relevant changes should be performed.

We have now altered vocabulary according to the suggestion, as controls were derived from two Serbian primary schools.

(c)   Additional information about the selection of controls (i.e., the case-control ratio, matching etc.) and the timeframe of the study should be reported.

We have now provided additional data about selection of the control group.

(d)  All the variables that were recorded should be provided. There should be no variables that are not reported in the Materials and Methods and are initially provided in other sections (such as the number of children).

We have now added additional data about all variables recorded.  

(e)   The rationale of modifying the questionnaire should be also reported. The authors should also provide data about the validity of the questionnaire after the modification.

The rationale of modyfying the questionnaire is now better described in text.

(f)    The section related to the informed consent should be modified accordingly if the parents gave the informed consent.

We are sorry for unprecised vocabulary, we have now changed the section accordingly.

(g)   The assessment of the continuous variables normality (normality tests) should be provided to report the results in mean and standard deviation values. If not normally distributed other measures of central tendency and dispersion should be used. For instance, to the best of my understanding, I guess that age should be not normally distributed. Based on that, the authors should also be cautious about the tests that will be used if, not normally distributed data is used.

Thank you for observation. We have now described statistical tests about normality of the data in Statistics subsection.

(2)   The last sentence (The MPVRQOL […] should be moved in the discussion since it is not related to the methodology used to perform this study.

Thank you for the suggestion. We have now moved the mentioned sentence in Disscusion section.

(3)   Finally, I strongly believe that a sample of the modified version of the questionnaire should be included as a figure since, to the best of my understanding, it serves as a novel modality that can be used by other authors interested in this field of research.

Thank you for your  suggestion. We have now provided the questionnaire translated in English as an Appendix.

  1. RESULTS

(1)   The content of the first paragraph (“The study […] (p>0.05)”.) should be reduced in size and a Table where all patients’ characteristics will be synopsized needs to be created. In addition, clauses such as “There were twice as many boys as girls […].” should be modified, to report only the findings of the study without commentaries.

We have now shortened the content of the first paragraph,and we have also created a new Table 1 where all summarized patients characteristics have been presented. We have also modified mentioned clauses in order to only report main findings of the study. 

(2)   I strongly believe that the term “gender” should be replaced by “sex”. Please refer to the following: https://www.coe.int/en/web/gender-matters/sex-and-gender

Thank you for acknowledging us of our oblivious mistake. The term “sex” is now replaced by word “gender” throughout whole manuscript. 

(3)   As mentioned in the Materials and Methods, all variables that are reported in the results (such as the largest number of children or the content of table 2) should be also included in the relevant section.

Thank you for your observation. We have now included all the reported variables in Methods.

(4)   The strength of the correlation coefficient as reported in lines 155-161 (“Statistical […] (Table 3).”) and in Table 3 should be explained in a greater extend using a reference where it is defined. 

The strength of correlation is now addresed by a refference inserted in Table 2 description. 

(5)   The content of the paragraph (“The questionnaire we used […] hoarse children.”) as stated in lines 165-186 should be significantly reduced in size. Since the relevant data is reported in a table, only a synopsis should be provided in the results and if needed, commentary can be performed in the discussion. Moreover, the authors should not repeat the structure of the questionnaire. 

We have now shortened the content of the paragraph, and avoided the repeating the structure of the questionnaire.

(6)   The content of the paragraph (“The largest number […] control group.”) should be significantly reduced in size. 

The content of the paragraph is now significantly reduced in size.

  1. DISCUSSION

Overall, I believe that it is an extremely well written and informative section. My biggest concern is only related to its length. Hence, I suggest being reduced as much as possible. Moreover, the study’s strengths and limitations, including the potential confounding and the bias assessment should be reported and discussed. 

Thank you for the comment. We have now reduced the length of the disscusion and also added one of possible limitations of the study.

As per the conclusion, I strongly believe that a separate section should be created. Yet, the text of this section should be reduced and, the authors should be more specific on addressing the novelty of this study as well as of the findings. 

We are obliged for the constructive proposition. We have now created a separate section for our conclusions. We have also pinpointed the main aspect of our study.